# What Is Injury in Ice Hockey: An Integrative Literature Review on Injury Rates, Injury Definition, and Athlete Exposure in Men’s Elite Ice Hockey

**DOI:** 10.3390/sports7110227

**Published:** 2019-10-23

**Authors:** Anthony S. Donskov, David Humphreys, James P. Dickey

**Affiliations:** 1Department of Kinesiology, University of Western Ontario, London, ON N6A 357, Canada; dhumphr4@uwo.ca (D.H.); jdickey@uwo.ca (J.P.D.); 2Donskov Strength & Conditioning, Columbus, OH 43229, USA

**Keywords:** sports injuries, injury surveillance, lower extremity injuries, professional hockey players

## Abstract

Injuries in men’s elite ice hockey have been studied over the past 40 years, however, there is a lack of consensus on definitions of both injury and athlete exposure. These inconsistencies compromise the reliability and comparability of the research. While many individual studies report injury rates in ice hockey, we are not aware of any literature reviews that have evaluated the definitions of injury and athlete exposure in men’s elite ice hockey. The purpose of this integrative review was to investigate the literature on hockey musculoskeletal injury to determine injury rates and synthesize information about the definitions of injury and athlete exposure. Injury rates varied from 13.8/1000 game athlete exposures to 121/1000 athlete exposures as measured by player-game hours. The majority of variability between studies is explained by differences in the definitions of both injury and athlete exposure. We were unable to find a consensus injury definition in elite ice hockey. In addition, we were unable to observe a consistent athlete exposure metric. We recommend that a consistent injury definition be adopted to evaluate injury risk in elite ice hockey. We recommend that injuries should be defined by a strict list that includes facial lacerations, dental injuries, and fractures. We also recommend that athlete exposure should be quantified using player-game hours.

## 1. Introduction

Ice hockey is a high intensity sport where players can reach speeds of up to 48 kph [1]. These speeds, and the nature of collision sports lead to musculoskeletal injuries at all levels of ice hockey [1,2,3]. There is a need to accurately quantify injury rates in men’s elite ice hockey both for assessing player risk [4] and the associated economic burden [5]. Injury rates in ice hockey have been investigated in order to assess injury trends, injury types, injury location, and underlying injury mechanisms [6]. Injury rates can also be used to quantify the effects of rule changes [7]. Accurate data is needed in order to better investigate areas of concern while objectifying the effects of rule changes and other preventative measures [8,9]. 

Differences in the definitions for injury and athlete exposure (AE) lead to inconsistencies between studies, and obscure the resulting injury rates. Consensus statements on injury definitions and data collection procedures have been developed for soccer [10] and rugby [11], but have not been developed for ice hockey. Consistent definitions and methods to evaluate ice hockey injuries are required [12] to improve the comparability of published data [8]. Our objective was to review global musculoskeletal injury rates in men’s elite ice hockey, as well as definitions of injury and athlete exposure. We focused our review on males as females have different types and rates of injury than males [13]. We focused on elite players aged 16 years and older playing junior hockey (United States Hockey League, North American Hockey League, Canadian Hockey League), US and Canadian College Hockey (NCAA Div. 1 and Div. III, Canadian Inter-University Sport), international or minor professional and professional hockey (Finnish Elite League, Swedish Elite League, Japanese Elite League, International Ice Hockey and the National Hockey League) as this cohort has not been as extensively studied as other levels such as high school and youth hockey [14,15]. In addition, the economic burden of injury at this level is high. During two seasons in the National Hockey League (NHL), injuries represented a total salary cost of US $218 million per year. While salary losses represent a significant financial burden, it is hoped that improved injury surveillance will reduce these costs. 

## 2. Materials and Methods

We conducted an integrative literature review [16] to evaluate musculoskeletal injury rates, injury definition and athlete exposure measurement in elite ice hockey. We formulated three research questions a priori to focus our review: What is the rate of musculoskeletal injuries in men’s elite ice hockey? In elite ice hockey, what injury definition is best suited to enable direct comparisons among research studies? Finally, in elite ice hockey, what measure of athlete exposure is best suited to achieve consistent and comparable injury rates? 

### Literature Search

A PubMed search strategy was created with the assistance of a University research librarian. PubMed was chosen as a search engine as it is the optimal tool in life sciences and biomedicine [17]. The search strategy used the key words: hockey AND (injury OR injuries) AND (NHL OR national OR international OR world OR competitive OR professional OR elite OR high caliber OR high caliber OR collegiate OR university OR intercollegiate OR NCAA OR “National Collegiate Athletic Association”). In addition, the same search strategy was performed on SPORTDiscus. The PubMed and SPORTDiscus records of these references were pooled and screened based on established inclusion and exclusion criteria (Table 1). Articles that were not relevant to our research questions were excluded. The references in the remaining papers were reviewed to identify additional relevant articles. All studies were reviewed by both authors for their relevance to the three research questions.

Original, peer-reviewed, English language research articles evaluating the injury rates in elite ice hockey were included. Articles were excluded if they were editorials, abstracts, books or excerpts from conference proceedings. Articles were excluded if they did not contain one of the following variables: injury definition, injury rate, athlete exposure, injury mechanism or injury location. Unpublished data was not reviewed. 

## 3. Results

The PubMed and SPORTDiscus search identified 2463 references. An additional 3 pertinent articles were identified from the references for these articles. A total of 2212 articles were vetted after 254 duplicate articles were removed. Two-thousand, one-hundred and eighty-four of these articles were excluded as they were not relevant to any of our three research questions. No relevant articles were published prior to 1975. Accordingly, a total of 28 articles were included (Figure 1). 

### 3.1. Rate of Musculoskeletal Injuries in Men’s Elite Ice Hockey (Question #1)

Injury rate data, and study design characteristics are presented for each of the 28 studies in Table 2. Injury rates in competitive ice hockey range from 13.8 to 121/1000 player-game hours, depending on factors such as the league of play and exposure estimate. Professional players in Europe and North America experience musculoskeletal injury rates between 49 to 80/1000 AE as measured in player-game hours [4,18] while the collegiate hockey players in Canada and the United States experience lower rates (13.8 to 19.95/1000 AE) as measured in player games [19,20]. The highest injury rates are experienced at the junior level (39.8 to 121/1000 player-game hours) [21,22]. The majority of these musculoskeletal injuries are attributed to collision with other players, the boards or the hockey puck [18,20,23,24]. 

The injury rates in practice are much lower than games. Practice rates range between 1.4/1000 player-practice hours for Swedish Elite hockey [30] to 3.9/1000 player-practice hours for junior hockey [21], versus game injury rates of 74.3/1000 player-game hours [38] and 121/1000 player-game hours [34], respectively. Although the injury rates are lower for practices, the number of hours spent in practices is several-fold greater than games, so the actual number of injuries is higher than indicated by the injury rate. 

Several long-term studies have assessed patterns in injury rates over time. For example, injury rates in the Finnish Elite League have increased from the 1970s (54/1000 AE) to the 1990s (83/1000 AE) using the player-game hours exposure estimate [35] (20 years). Overall game injury rates increased 1.8% annually over a seven-year period (2000–2007) in men’s NCAA ice hockey using the player game estimate. Practice rates also increased 7.8% annually during this time [39]. In contrast, on average, injury rates have decreased between 2007 and 2013 in men’s International Ice Hockey Federation World Championship tournaments [42] (6-years). One Canadian Intercollegiate team also experienced decreases in injury rate over a six-year period from 11.3 to 8.30/1000 player games (1991–1996) [37] (6-years). 

There was a large variance in injury rates between studies. This large variance is a function of variability in the definitions for both injury and athlete exposure. As noted in previous papers, establishing consistent definitions of injury and athlete exposure are important first steps for objectifying injury risks in high caliber ice hockey [10,45]. 

### 3.2. Injury Definition in Men’s Elite Ice Hockey (Question #2)

Probably the most important methodological factor affecting injury rate calculations is the definition of what constitutes an injury [45]. A review investigating the methods of data collection on injury surveillance identified three categories of injury definitions [45]. Category 1 defines injuries as all complaints regardless of time loss. All injuries are recorded, regardless of the severity or amount of time lost from competition. Category 2 defines injuries as events that require medical attention by a member of the medical staff. Therefore, according to this definition, a member of the medical staff, typically a team therapist or team doctor, must diagnose the injury. Finally, category 3 defines injuries as events that have a time loss element. Accordingly, an injury is only recorded if the athlete misses a team-related practice or competition. Individual studies typically fit into one, or more of these categories.

Our review identified 28 studies evaluating injuries in elite ice hockey. Early research investigating injury rates in the Swedish Elite League, and the Swedish National team used the time loss definition of injury (Category 3). As shown in Table 2, the majority of ice hockey injuries studies use either a time loss (Category 3) or medical attention definition (Category 2). None of the articles evaluating injuries in elite ice hockey used the all complaints definition (Category 1). 

Our review found inconsistent definitions of a reportable injury in ice hockey research based on the time loss definition. In addition, the list of injuries has expanded over time. Facial lacerations were considered reportable injuries in 1991 [24], while sutures, fractures, dislocations and subluxations were added in 1992 [23]. Concussions, dental and eye injuries were added in subsequent years [20,21], potentially increasing injury rates by expanding the list of injuries. In addition, illness may be counted as an injury, inflating the injury rates [23]. 

The definition of injury based on medical attention (Category 2) has also been used to quantify competitive ice hockey injury rates [31,34,38]. However, this metric is often combined with the time loss component to result in a broader interpretation of injuries [20,21,22,23,35,42]. For example, injuries such as concussions, dental injuries, lacerations and eye injuries are captured with medical attention by a team physician or athletic trainer, resulting in a more extensive list of ice hockey related injuries compared to definitions that did not include these injuries [45]. Of note, some studies have expanded their list to include illnesses and psychological complaints that are unrelated to injury [46]. 

The time loss definition (Category 3) is the easiest to use as it is easy to track time loss. However, it leads to the fewest reported incidents [45] as it fails to capture the athletes that continue to train and play while injured [47]. Depending on the time of year, some injuries may be under reported as injured players continue to play throughout key time periods, such as playoffs. The medical attention definition (Category 2), though broader and encompassing a greater number of conditions, also has limitations. The subjective interpretation of what constitutes medical attention may lead to systemic bias [48], and the types of injuries managed by the various practitioners may differ based on their qualifications and status [45].

### 3.3. Athlete Exposure Metric in Men’s Elite Ice Hockey (Question #3)

Athlete exposure is the second component of injury rate. An athlete exposure is defined as one athlete participating in a practice or game in which there is a potential for athletic injury [49]. Injury rates are typically based on 1000 athlete exposures. These exposure rates can be quantified as injuries per 1000 game-hours (or injuries per 1000 games), injuries per 1000 practice-hours, or overall injuries per 1000 AEs (games and practices combined). Injury per 1000 player-game hours is based on a 60-min active game and is calculated as the number of injuries/number of players on the ice at the same time (6)/number of games × 1000. Many researchers use this method [18,21,24,30,31,33,34]. However, this exposure estimate is not used consistently among researchers. For example, several studies accounted for both teams when calculating athlete exposure (number of injuries/number of players on ice at the same time (two teams)/number of games × 1000 [22,42]. In contrast, another study used a 20 person roster, including the back-up goaltender, to calculate athlete exposure per 1000 player-game hours [38]. This larger number of players will lead to a smaller injury rate. 

Our review identified different nomenclatures pertaining to the athlete exposure metric, such as player-games and player-game hours [42]. The number of athletes used to quantify these exposure rates vary between studies, and are not consistently defined. For example, one researcher [42] calculated player-game injury rates based on 22 players competing for each team in a game (i.e., 44 players) while another [30] calculated player-game hours injury rates based on 6 players. This was based on the number of players on the ice at a time, and whether goaltenders were included. Other researchers have used roster averages over a set period of time [36,37], or a tournament [22,42] to calculate player-game injury rates. 

Injury per 1000 games is the average number of injuries that one player experiences per 1000 games (number of injuries/total number of players (roster)/number of games × 1000 [20,37]. Our review found different implementations of this approach as there was some research that counted both rosters when computing athlete exposure [42]. This has an effect on total estimated exposures and can lead to reduced injury rates. Finally, several articles did not fully describe whether they included both rosters or a single team roster when calculating athlete exposures [19,23], making it difficult to determine accurate injury rates. 

In addition, we investigated the impact of calculating injury rate based on the actual time on ice (TOI) [4,50]. Using the actual time on ice, injury rate was calculated as the number of injury events/sum of individual AE time as found on the player statistics page (www.nhl.com/stats/player). The time on ice was calculated based on the number of minutes and seconds that each individual played per game over the season. The difference between estimated athlete exposure (number of injuries/number of teams (30)/number of players on roster each game (19)/number of games (82)) and the TOI metric was large. As much as three times the amount of exposure was identified by estimating exposure rates. However, when comparing the time on ice metric to the estimated player game-hour metric, the differences were minimal. The player game-hour exposure (based on one hour per game rather than the actual amount of time that players spent on ice, which changes due to overtime periods and penalties) is similar to the time on ice calculations (14,676.2 h calculated as the sum of players’ time on ice versus 14,760 h calculated as 30 teams × 82 games × 6 players) [4].

Our review found that practice athlete exposure was calculated consistently in most studies. Injury per 1000 practice hours (number of injuries/number of practice hours/number of players on team × 1000) was the standard [21,30,33,34]. 

## 4. Discussion

Injury rates in men’s elite ice hockey are higher in professional leagues such as the Swedish Elite League [31] and Finnish National League [33] than college hockey [19,20,23]. This may be due to the differing demands as professional players play more games in a season, and therefore may experience more overuse injuries. It may also be due to the athlete exposure estimation (player-game hours vs. player-games) used to calculate injury rate. Style of play and hockey rink dimensions are additional variables that may influence injury rate. Overall, we observed the trend that injury rates have increased over time in professional European leagues [35] and college hockey [39], while decreasing in men’s international ice hockey [42].

We observed a wide range of injury definitions. This affects both the reliability and comparability of injury surveillance research. There is currently a consensus-based injury definition in sports such as soccer and rugby [10,11]; however, there is no consensus injury definition in ice hockey. We recommend that hockey forms a consensus injury definition as this will resolve an important issue that currently impedes hockey injury research. A consistent injury definition would create clarity as to which injury is considered a recordable event. We identified the International Ice Hockey Federation’s (IIHF) definition of injury as the most appropriate as it only captures events that are sufficiently severe that they influence participation in practices or games. The IIHF’s definition describes a reportable event as “any injury sustained in a practice or game that prevented the player from returning to the same practice or game; any injury sustained in a practice or game that caused the player to miss a subsequent practice or game; a laceration which required medical attention; all dental injuries; all concussions; all fractures” [42]. Although no single definition suits all needs, the time loss definition is the most common and easy to identify. It is considered reliable and allows for the comparison of data between teams, seasons and various leagues [45]. It is also used in other professional sports such as cricket and Australian football [51,52]. The choice of definition should reflect the aims and goals of surveillance. With its consistency, ease of use, and comparability of published data [8] among the most important variables, we feel the time-loss definition best meets the needs of injury surveillance in men’s elite ice hockey. However, like all definitions there are limitations in choosing this metric. First, athletes often continue to compete in the presence of injury. Delaying treatment may lead to missed injuries. Finally, the threshold for time loss may depend on the time of season and how important the player is to team success [45]. Despite these drawbacks, we feel the strengths of the time-loss definition outweigh its limitations and that the IIHF’s time-loss definition is warranted in elite men’s ice hockey. 

We also noted that athlete exposure estimations were inconsistent in the literature. The major confusion lies in how many participants are included in the injury rate calculation. Several researchers used player-game exposure based on the entire team, or average team roster (19 players) [20,36,37], while others used player-game hour exposures based on 6 players [18,21,24,30,34]. This leads to difficulty in interpreting injury rates and comparing research. It was proposed that the gold standard in athlete exposure during games is time on ice. As much as three times the amount of exposure was accounted for by estimating exposure rates using the player-game approximation compared to time on ice. However, when comparing the time on ice metric to the estimated player game-hour metric (based on one hour per game, rather than the actual amount of time that players spent on ice) it appears that this difference is small [4]. Therefore, the simplest and easiest way to calculate athlete exposure is to use six players on the ice (player-game hours) unless position specific injury rate information is warranted. Using a consistent athlete exposure metric will increase intra- and inter-league injury rate reliability. 

The majority of studies reviewed found that collision with other players is the leading mechanism of injury as well as contact with the boards, opponent’s hockey sticks and hockey pucks [22,35,36]. This leads to an injury paradox: the goal of the sports performance specialist is to build bigger, faster, stronger, leaner, more powerful, robust players. However, these types of players also travel faster, and hit harder, elevating the risk of injury. This situation emphasizes the need for accurate injury surveillance methods as these may help reinforce rules and/or govern the addition of new rules enforcing safety for active players.

### Limitations 

There are limitations to this study. There is a relative paucity of studies evaluating injury rates in men’s elite ice hockey, and the definitions of injury and athlete exposures vary between studies. Accordingly, the reported injury rates differ between studies and are difficult to interpret. Two databases (PubMed and SPORTDiscus) were used to identify research papers that were relevant to injury definition, injury rates and athlete exposure in elite ice hockey. While these databases are an excellent source for research articles in sports, life sciences and biomedicine, supplemental databases may have identified additional research studies. 

## 5. Conclusions

In summary, this project represents the first integrative literature review investigating injury rates, injury definition and AE in men’s elite ice hockey. The greatest opportunities for continued improvement lie in both consistency and comparability to refine, improve and streamline calculations of injury rate. 

At the current moment, a uniform definition of injury is the most important step to better objectify injury data in ice hockey. A universal definition is required by sport governing bodies and researchers. Though each approach has its limitations, in order to compare exposure rates in both the intra- and inter-league, a workable, consistent definition is required. Specific responsibility should be given in terms of who will diagnose the injury if the definition is a time loss definition, a medical attention definition, or a combination. In addition, a detailed injury list is needed to clarify the definition of injury and whether specific injuries such as dental, concussions, and facial lacerations, are included. 

Finally, disparate AE estimations diminish injury rates, which compromises research findings. Attendance rate in both practice and games (player-game hours based on 6 players per game and the full roster during practices) is the preferred method for calculating athlete exposure. 

### Further Research

Investigating anatomical areas prone to injury is crucial for team performance staff such as athletic therapists, physical therapists and strength and conditioning specialists as it may guide rehabilitation initiatives, performance program design and athlete monitoring [53]. We observed that the lower extremities was the most common site of musculoskeletal injury.

Future research should clearly define injury rate measurements to provide doctors, therapists, and coaches with accurate information to streamline return to play initiatives. In this regard, our review has exposed gaps including the disparate definition of injury and the lack of a consistent athlete exposure metric.

## Figures and Tables

**Figure 1 sports-07-00227-f001:**
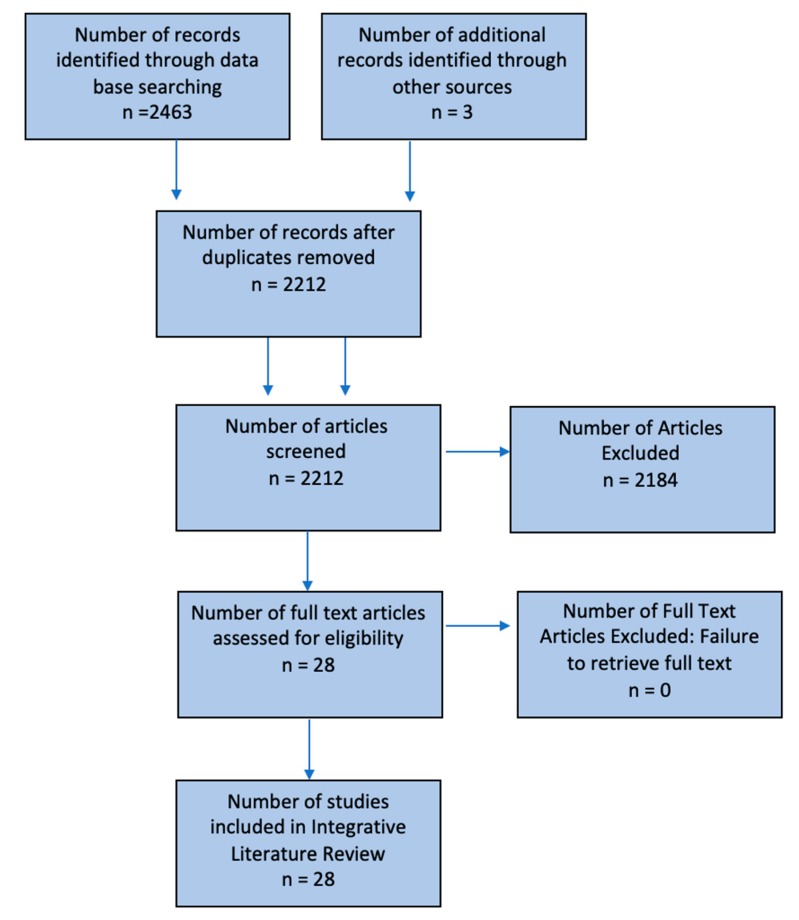
Flowchart describing the process for selecting relevant studies. The top row represents the identification process. The second and third rows represent the screening process. The fourth row represents the eligibility of the articles assessed and the last row identifies the articles included.

**Table 1 sports-07-00227-t001:** Inclusion/exclusion criteria for literature search.

Factor	Inclusion Criteria	Exclusion Criteria	Rationale for This Criterion
Publication Type	Peer-reviewed original research articles only	Review papers, non-peer reviewed articles, editorials, abstracts, book chapters and conference proceedings	For practical reasons, it was deemed to exclusively review primary research articles, rather than non-peer reviewed or abbreviated sources.
Language	English language	Non-English	For practical reasons, it was deemed acceptable to only include studies published in English.
Publication Date	November 1976 to April 2019	Publications prior to January 1975	The characteristics of ice hockey injury reporting may change over time due to rule changes, technological advancements and education. Literature was captured backdated to 1988 to capture these potential developments.
Study Design	Multi-center studies, randomized control trials, cohort studies, case-controlled studies and cross-sectional studies.	Case studies	Study design was chosen to ensure reasonable empirical support, and high methodological rigor in defining injury and injury rates amongst competitive hockey players.
Gender and Age	Men athletes aged > 16-years participating in a competitive league/team	Women only studies or men ages < 16, age unspecified involved in youth sport	The primary outcome of interest was injury definition and injury rate calculation in competitive ice hockey played by men. Studies that compared rates between males and females and have separate data for both genders were also included for baseline comparisons. Men athletes aged > 16 were considered appropriate. This age demographic represents elite players.
Playing Level	Competitive participation	Recreational sport/training	The primary outcomes of interest are injury definition, injury rates, mechanism and anatomical location sustained during competitive ice hockey.
Sport	Injuries must be sustained during ice hockey games and practices	Any sport other than ice hockey	Sports included other than ice hockey may result in definitions, and injury rates that are too broad.
Types of Injury	Injuries to the musculoskeletal system, including strains, sprains, breaks	Concussions, spinal injuries, head/face, lacerations	The primary outcomes of interest are soft tissue injuries of the upper and lower extremity during competitive ice hockey
Outcome Measures	Injury definition, injury rates, athlete exposure, mechanisms, anatomical location	Outcomes other than injury definition, injury rate, and athlete exposure, mechanisms and anatomical location	The primary outcomes of interest are injury definition, injury rates, mechanisms and anatomical location.

All studies were reviewed by both authors. The study methodology, and outcome measures were extracted.

**Table 2 sports-07-00227-t002:** Summary of papers evaluating injury definition, injury rate, athlete exposure and injury mechanism in men’s elite ice hockey.

Authors	Year	Demographic	Injury Definition	Type	Injury Rate	Mechanism of Injury	Injury Type	Injury Rate Computation
Hayes [25]	1975	Intercollegiate Ice hockey	“An event requiring some attention by the team trainer or physician or both.”	Medical Attention	1.14 injuries per game (Canada) 1.28 injuries per game (USA)	Collision	Head and face, knee, shoulders	Total injuries/Total number of games
Sutherland [26]	1976	Youth-Pro	The injuries were classified according to the standard nomenclature of athletic injuries as recommended by the American Medical Association	N/A	Pro Group: 143/1000 AE (practice and games)	N/A	Scalp and face 60.8%, Groin 9.1%, knee 7.8%, shoulder 5.9%	N/A
Hayes [27]	1978	Youth-Pro	“Any change in the normal, healthy state of the individual that requires medical attention and disables a player either temporarily or permanently.”	Medical Attention	University: 1.17/Game Professional: 1.15/Game	Stick and puck contact	Contusions and lacerations	Total injuries/Total number of games
Rielly [28]	1982	College Hockey	“A reportable injury was defined as being one that required definitive physical evaluation and medical treatment.”	Medical Attention Definition	1/12.7 h of play **	Player contact (43.3%), puck contact 27%	Face, hips, shoulders	N/A
Meeuwisse et al. [29]	1988	Canadian University	Injury was defined as any disability arising either in practice or competition that required physical attention.	Medical Attention Definition	As calculated by percentage. Hockey had the greatest percentage of players injured.	N/A	Knee, ribs, low back	N/A
Lorenzton, Wedren, Pietila [30]	1988	Swedish Elite Team	“Injury was defined as any injury occurring during on-ice practice or games and causing the player to miss the next practice session or game. Facial lacerations, which are common in ice hockey but do not cause absence from practice or game are reported separately.”	Time Loss Definition	78.4/1000 player game hours, 1.4/1000 player practice hours	Checking 32.9%, Player contact 25%, Puck contact 14.5%, stick contact 11.8%, collision with boards 6.6%, cutting, 6.6%, skate contact 2.6%	Contusions, strains and sprains were the most common types of injury. Knees were the most commonly injured joint (5 injuries were complete tears of the MCL). 53.7% of injuries were localized in the lower limb.	Practice Injury Rate = number of injuries/Practice Hours × Roster (23.4). Games Injury Rate = number of injuries/Game hours × Total Players on ice (6).
Lorenzton, Wedren, Pietila, Gustavsson [18]	1988	Swedish National Team (40 International games)	“Injury was defined as any injury occurring during games and causing the player to miss the next practice session or game. Facial lacerations, which are common in ice hockey but do not cause absence from practice or game are reported separately.”	Time Loss Definition	79.2/1000 player game hours	Player contact 42.1%, checking 31.6%, collision with boards 10.5%, puck/skate contact 5.3%	Contusions, strains and sprains were the most common types of injury. Knees were the most commonly injured joint, followed by the thigh and wrist.	Total Injuries/Total AE × 1000; Total AE = games × Total players on ice (6)
Tegner, Lorentzon [24]	1991	Swedish Elite League (12 teams)	“Injury was defined as any injury occurring during ice practices or games and causing the player to stop playing or to miss the next practice session or game. Facial lacerations, which are common in ice hockey, but do not cause absence from practice or games, are also reported.”	Time Loss Definition	53/1000 player game hours (76% of injuries occurred during games)	Stick contact 25.5%, player contact 24%, puck contact 11.2%, collision with boards or goal posts 9.7%	Strain, laceration and contusions were the most common types of injury. Knees were the most common joint injured (13.2%), followed by the hip (12.1%)	Total Injuries/Total AE × 1000 Total AE = games × Total players on ice (6)
McKnight, Ferrara, Czerwinska [23]	1992	Collegiate (Div. I)	(1). Loss of practice or game time because of injury/illness, (2). Injury that required sutures even if no time loss was involved, (3). Injury in which a fracture or dislocation/subluxation occurred even if the athlete was able to continue participation	Time Loss/Medical Attention Definition	Total: 10.22/1000 AE Games: 14.73/1000 game hours. Practice: 2.52/1000 practice hours	Person/Ice Impact 42%, impact with the boards 32%. The shoulder and knee had the highest rate of injury when compared to other body parts	Contusions and strains were the most common types of injury	number of Injuries/Total AE × 1000 (games and practice)
Pelletier, Montelpare, Stark [20]	1993	Canadian Inter-collegiate	“Any brain concussion causing cessation of the athlete’s participation for physical observation before return to play, any dental injury requiring professional attention, any injury/illness causing cessation of an athlete’s customary participation throughout the participation day following day of onset, or any injury/illness requiring substantive professional attention before the athlete’s return to competition.”	Time Loss/Medical Attention Definition	19.95/1000 AE (player games)	Body checking 44.6%, collision (accidental) 28.8%, stick 12.2%, fighting 6.5%, illegal body check 5.8%, non-contact 2.2%	Sprains (31%) and contusions (21%) were the most common type of injury. Knees were most frequently injured (18.6%), followed by teeth and eyes (17.6%), and shoulders (14.9%),	Total Injuries/Total AE × 1000 Total AE = games × Total players on team (19)
Pettersson, Lorentzon [31]	1993	Swedish Elite League	“Injury was defined as any injury occurring during on-ice practice or games and requiring medical attention and treatment. Injuries causing the player to miss the next practice or game have been analyzed separately.”	Medical Attention Definition	74.1/1000 game hours	Stick contact 26.1%, player contact 23.9%, puck contact 16%, collision with boards 7.2%, fall (no contact) 4%	Contusions, lacerations, sprains and strains are the most common mechanisms of injury. Knees were the most common joint injured followed by the thigh, groin and shoulder	Total Injuries/Total AE × 1000 Total AE = games × Total players on ice (6)
Stuart, Smith [21]	1995	United States Hockey League	“Injury was defined as an event that kept a player out of practice or competition for 24 h, required the attention of a team physician (e.g., suturing lacerations) and included all dental, eye and nerve injuries and concussions.”	Time Loss/Medical Attention Definition	Overall injury rate was 9.4/1000 player hours, game injury rate was 96.1/1000 player hours, practice injury rate was 3.9/1000 player hours	Collision 51%, stick contact 14%, skate/puck contact 11%, off-ice injuries 8%	Strains, lacerations and contusions were most common mechanism. Aside from the face, the shoulder, hip, lumbar spine and knee were the most common anatomical sites of injury	Total Injury/AE × 1000 = Practice Injury Rate; Practice AE = Practice Hours × Roster (25) Total injuries/Total AE × 1000; Total AE = number of games × Total players on ice (6)
Cunningham [32]	1996	University Games	“A recordable injury was defined as any incident occurring during warm-up or competition and which required medical attention, on-field management to enable continued participation, or removal from the playing field.”	Medical Attention	33.5% of injuries in relation to total number playing the sport	Player collision	Muscle strains and hematoma (21.7%)	Number of injuries/number of players participating
Molsa, Airaksinen, Nasman, Torstila [33]	1997	Finnish National League, Finnish First Division	“An injury was defined as any trauma occurring during practices or games and causing absence from the next practice or game or needing treatment (ex. stitches), examination by a physician (ex. radiographs), or rehabilitation prescribed by a physician (ex. physical therapy). Injuries due to overuse were excluded.”	Time Loss/Medical Attention Definition	66/1000 player-game hours, 36/1000 player game hours (Div. I)	Checking 29.7%, stick 14.6%, contact with opponent 14.6%, puck 7.9%	Contusions, strains and sprains were the most common type of injury, the knee joint and groin were the most common locations	Total Injury/AE × 1000 = Practice Injury Rate Practice AE = Practice Hours × Roster (21) Total Injuries/Total AE × 1000; Total AE = number of games × Total players on ice (6)
Pinto, Kuhn, Greenfield, Hawkins [34]	1999	Junior A Hokey Players (22 players)	“An injury was defined as any event that required the attention of a physician or trainer.”	Medical Attention Definition	121/1000 player game hours	Contact with stick 16.2%, overuse 13.5%	Sprains/subluxations/dislocations were the most common, aside from the face, the shoulder and knee were the most common	Total Injury/AE × 1000 = Practice Injury Rate; Practice AE = Practice Hours x Roster (22) Total Injuries/Total AE x1000 Total AE = #games × Total players on ice (6)
Molsa, Kujala, Nasman, Lehtipuu, Airaksinen [35]	2000	Finnish Elite League (7 teams, 3 different decades)	“An injury was defined as any sudden trauma occurring during practice or game that led to examination and treatment by a physician.” Minor injuries requiring no absence were also included, but minor injuries needing no medical care and injuries due to overuse were excluded	Medical Attention Definition	Game injury rate increased from 54/1000 player hours in the 70’s to 83/1000 player hours in the 90’s, most common mechanism was collision	Checking, stick, falling, collision with opponent, puck, collision with boards	Contusions, sprains/strains and lacerations were the most common mechanisms of injury. The knee was the most common major injury of the lower quadrant	Player years of exposure, (Seasons × Teams × Players) × Practice Hours x Roster = Practice Injury Rate: Player years of exposure, (Seasons × Teams × Players) × Game Hours × Roster (6) = Game Injury Rate.
Flik, Lyman, Marx, [19]	2005	American Men’s Collegiate Ice Hockey (8 teams/1 season)	“An injury was defined specifically as any injurious episode that led to loss of participation in the immediate subsequent AE, whether it was a practice or game.”	Time Loss Definition	Overall injury rate was 4.9/1000 AE, 13.8/1000 AE games, 2.2/1000 AE practice	Collision with opponent 32.8%, collision with boards 18.6%, overuse 8%, puck 6.2%	Concussions were the most common, followed by knee (MCL) and shoulder injuries	Total Injury/AE × 1000 = Practice Injury Rate Practice AE = Practice Hours × Roster Est Total Injuries/Total AE × 1000; Total AE = number of games × Total player avg attendance
Agel, Dompier, Dick, Marshall [36]	2007	NCAA Men’s Ice Hockey (16 years of data: Div. I-III)	“A reportable injury in the ISS was defined as one that (1) occurred as a result of participation in an organized intercollegiate practice or competition and (2) required medical attention by a team certified athletic trainer or physician and (3) resulted in restriction of the student-athlete’s participation or performance for 1 or more calendar days beyond the day of injury. The injury definition was expanded in the ‘94–95’ academic year to include any dental injury occurring in an organized practice or game, regardless of time lost.”	Time Loss Definition	16.27/1000 AE games, 1.96/1000 AE practice	Player contact 50%, other contact 39.6%, no contact 9.7% (game numbers). Injury was 8x higher in games.	Knee internal derangement (13.5%) was the most common lower extremity injury reported during games, followed by concussions and AC injuries. Whereas pelvis and hip strains (13.1%) were the most common injury reported at practice.	Total Injury/AE × 1000 = Practice Injury Rate; Practice AE = Practice Hours × Roster (26) Total Injuries/Total AE × 1000 Total AE = number of games × Total players (19)
Rishiraj, Lloyd-Smith, Lorenz, Michel [37]	2009	Men’s Varsity Ice Hockey (Canada)	“Any event, during team or team related game, practice, and/or activity (on or off the ice), requiring any attention by the team’s therapist and/or physician and subsequent game and/or practice time loss.”	Time Loss Definition	3.7/1000 player game and practice exposure	Non-contact, ice/board contact, body contact	Sprains 20%, strains 20%, concussions 13% and contusions 12%	Total Injury/AE × 1000; Practice AE = Practice Hours × Roster Total Injuries/Total AE x1000 Total AE = number of games × Roster Selected
Kuzuhara, Shimamoto, Mase [38]	2009	Japanese Elite Team	“An injury was defined as any event that occurred during on-ice practices or games that required medical attention and treatment.”	Medical Attention Definition	74.3/1000 player game hours, 11.7/1000 player-game hours for injuries resulting in any time loss, 11.2/1000 player-practice hours, 1.1/1000 player-practice hours for injuries resulting in any time loss	Overuse 52%, puck contact 21%, stick contact 15%, falling 12%	Contusions 35.4%, strains 15.6%, lacerations 9.3%	Overall injury rate (regardless of time loss): #of injuries/number of hours per 1000 player-hours number of injuries causing time loss (>1 day)/number of hours per 1000 player-hours Number of player: 2003 (20 players/game, 25 players/practice), 2004 (20 players/game, 37players/practice), 2005 (22 players/game, 32 players/practice)
Agel, Harvey [39]	2010	NCAA Men’s and Women’s Ice Hockey (Div. I and III)	Same as Dick et al. above	Time Loss Definition	18.69/1000 AE games, 2.23/1000 AE practice for men, 12.10/1000 AE games, 2.90/1000 AE practice for women	Player contact 48%	The most common injury among men was concussion followed by shoulder and knee ligamentous in juries	Number of Injuries/Number of AE (games or practice × roster) × 1000
Engebretsen, Steffen, Alonso, Dvorak, Junge, Meeuwisse, Mountjoy, Renstrom, Wlikinson [40]	2010	Olympic Sport	“An athlete was defined as injured or ill if he/she received medical attention regardless of the consequences with respect to absence from competition or training.”	Medical Attention Definition	A total injury rate of 111.8/1000 AE was reported for both males and females. A total of 276 males were registered with 44 total injuries (16%) in men’s elite ice hockey.	N/A	N/A	Number of Injuries/Athlete Exposure
Kerr, Dompier, Snook, Marshall, Klossner, Hainline, Corlette [41]	2014	NCAA Sports	“Any injury occurring during an organized intercollegiate practice or game.” (1982) “A reportable injury was defined as an injury that (1) occurred as a result of participation in an organized intercollegiate practice or competition, (2) required attention from an AT or physician, and (3) resulted in restriction of the student-athlete’s participation for 1 or more days beyond the day of injury.” Multiple injuries from one event could be included. In addition, AT’s were asked to include any dental injuries that occurred in an organized practice or game, regardless of time lost. 2003-onward). Beginning in 2009–2010 academic year, non-time loss injuries were also monitored.	Time Loss/Medical Attention Definition	N/A	N/A	N/A	Number of Injuries/AE (average team roster) × 1000
McKay, Tufts, Shaffer, Meeuwisse [4]	2014	NHL Players (2006–2012)	“Any event captured by the IIE form, and restricted to those designated as practice-related or game related, resulting in one or more-man games lost.	Time Loss Definition	15.6/1000 AE based on estimated AE’s. Based on recorded TOI *, the injury rates were roughly threefold higher at 49.4/1000 player-game hours	Body checking was the most common mechanism	Most commonly injured body regions were the head (16.8%), thigh (14%), and knee (13%)	Estimated AEs = 82 games × 19 players (including goalie) TOI (NHL.com) = number of injury events/sum of individual AE time
Tuominen, Stuart, Aubry, Kannus Parkkari [42]	2015	Men’s International Ice Hockey (2006–2013)	“The definition of an injury was made in accordance with the accepted international ice hockey norms: (1) Any injury sustained in a practice or a game that prevented the player from returning to the same practice or game, (2) any injury sustained in a practice or a game that caused the player to miss a subsequent practice or game, (3) a laceration that required medical attention, (4) all dental injuries, (5) all concussions, (6) all fractures	Time Loss/Medical Attention Definition	14.2/1000 AE player games, 52.1/1000 AE player game hours For WC A-pool tournaments and Olympic games the injury rate was 16.3/1000 player-games, 69.6/1000 player-game hours	Body contact and puck contact were the mechanisms	Most common types of injuries were lacerations, sprains, strains, and contusions. The knee was the most commonly injured lower body segment, MCL was the most common, and the shoulder was the most common site of an upper body injury.	Player game injury rate (based on 22 players on each team): Number of injuries/number of players (two teams)/number of games × 1000, Player game-hour injury rate (based on 6 players on ice at once): number of injuries/number of players on ice at the same time (two teams)/number of games × 1000
Kerr et al. [43]	2015	NCAA Ice Hockey	Injuries were defined as those that occurred in an organized NCAA-approved practice or competition and required medical attention by a physician or athletic trainer. An athlete-exposure was defined as one student-athlete’s participation in one practice or one competition.	Medical Attention Definition	9.5/1000 AE	N/A	Concussions, contusions, fractures	Number of Injuries/Number of Athlete Exposures
Tuominen, Stuart, Aubry, Kannus, Parkkari [22]	2016	World Junior Hockey Players (ages 18–20)	“The definition of an injury was made in accordance with the accepted international ice hockey norms: (1) Any injury sustained in a practice or a game that prevented the player from returning to the same practice or game, (2) any injury sustained in a practice or a game that caused the player to miss a subsequent practice or game, (3) a laceration that required medical attention, (4) all dental injuries, (5) all concussions, (6) all fractures	Time Loss/Medical Attention Definition	11/1000 AE player-games, 39.8/1000 player-game hours	Body checking 32%, stick 13%, and puck contact 13%	The knee was the most frequent site of lower body injury in WJ and U20 tournaments (33%), MCL sprain was most common, the shoulder was the most common upper body injury.	Player game injury rate (based on 20–22 players on each team): number of injuries/number of players (two teams)/number of games × 1000, Player game-hour injury rate (based on 6 players on ice at once): number of injuries/number of players on ice at the same time (two teams)/number of games × 1000
Lynall, Mihalik, Pierpoint, Currie, Knowles, Wasserman, Dompier, Comstock, Marshall, Kerr [44]	2018	Collegiate Men’s and Women’s Hockey (2004–2005, 2013–2014)	“An injury that (1) occurred as a result of participation in an organized practice or competition; (2) required medical attention by a certified AT or physician; and (3) resulted in restriction of the student-athlete’s participation for 1 or more days beyond the day of injury. Since the 2007–2008 academic year, HS RIO has also captured all concussions, fractures, and dental injuries, regardless of time loss.” “Beginning in the 2009–2010 academic year, the NCAA-ISP also began to monitor all non–time-loss injuries. A non–time-loss injury was defined as any injury that was evaluated or treated (or both) by an AT or physician but did not result in restriction from participation beyond the day of injury.”	Medical Attention/Time Loss Definition	Collegiate Men: 13.45/1000 AE	Collision	Strains/Sprains	Total Injuries/Total Athlete Exposure

TOI * = Time on Ice. ** = Author did not specify how injury rate was calculated.

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
