# Peer review of "What Is Injury in Ice Hockey: An Integrative Literature Review on Injury Rates, Injury Definition, and Athlete Exposure in Men’s Elite Ice Hockey"

_sports, 2019, doi:10.3390/sports7110227_

Round 1
Reviewer 1 Report
Thank you for the opportunity to review the manuscript titled “What is Injury In Ice Hockey: An Integrative Literature Review on Injury Rates, Injury Definition, and Athlete Exposure in High Caliber Ice Hockey”. The authors are to be commended for attempting to conduct an integrative review on injuries in ice hockey. However, there are several methodological choices that potentially diminish the value of this integrative review. Please see specific comments below.
For an integrative review purporting to evaluate and critique the literature on injury in ice hockey, with a view to improving future research in the field in terms of improved consistency in the use injury and exposure definitions, it is unfortunate that the authors have decided to limit their review to musculoskeletal injury. I agree that more consistency and consensus of operational definitions and surveillance methods in ice hockey would be very useful, but this would be best achieved if a more comprehensive approach was taken. That is, an approach more in line with previous efforts to achieve consensus in soccer, rugby, track and field, aquatic sports, cricket, and tennis. This value of this review is severely diminished by the fact it does not account for non-musculoskeletal limb injuries (i.e. concussion, spinal injury, and head/face injury). Similarly, why restrict the review to “high caliber” men’s hockey? Why exclude female and lower level athletes? Surely, the issues with lack of consistency and consensus about operational injury definitions and surveillance methods are just as relevant for female and non-professional ice hockey. The literature appears to have been very restrictive. This is not only suggested by the listed search terms and the fact that PubMed returned only 106 hits, but the fact that almost two-thirds of the included articles were found through hand-searching the bibliographies of included studies. The objective of this review, as provided in the Introduction section (lines 40-42) and the Methods section (Table 1), are inconsistent. Whereas the former refers to “lower-extremity musculoskeletal injury”, the latter refers to “injuries of the upper and lower extremity”. I find it somewhat ironic that the authors discuss issues around the variability in the operational definition of exposure and reporting of injury rates (Section 3 of the Results), but nevertheless conflate many of these as athlete-exposure (AE) in their preceding section on injury rates (Section 1 of the Results). For instance, in the first paragraph (lines92-96), the authors refer to a practice [injury] rate of 1.4/1000 AE, where the denominator is in fact 1000 player-practice hours. There section has several other examples where the authors report rates per 1000 AE, when the rates actually incorporate (estimated) exposure time. The Results and Discussion focussing on specific anatomical areas feels like a slight of hand. While the Results and Discussion about injury rates pertain to musculoskeletal injuries of upper and lower limbs, the Results and Discussion on specific anatomical areas pertain to musculoskeletal injuries of the lower limb only. The reason for the switch from both upper and lower limb injuries to lower limb injury only is not made clear. Moreover, the authors appear to have fallen prey to their own slight of hand when they conclude that their review has exposed the prevalence of knee and hip injuries in ice hockey. Firstly, the authors have not reported any prevalence data, they have merely summarised the proportion of knee and hip injuries among lower limb injuries. It should not come as a surprise that the knee and hip comprise a large proportion of lower limb injuries, but that does not entail that the prevalence of knee and hip injuries is high or that the knee and hip comprise a large proportion of all injuries. In the Discussion section, the authors assert that the International Ice Hockey Federation’s injury definition is the most appropriate “as it only captures event that are sufficiently severe that they influence participation in practice or games”. However, this assertion is without justification or argument. What does sufficiently severe even mean? How is “influence on participation in practice or games” defined and measured in the first place? Complete time-loss? What about injuries that result in activity modification, but no time-loss? What about injuries that require treatment, but does not result in time-loss (with or without activity modification)? Why is it more appropriate to not capture such injuries? Surely, what is more appropriate will depend on what the measured injury rate is used for. For instance, estimating the true risk or burden of injury will require a different operational injury definition compared to estimating the necessary resource allocation for medical personnel in an ice hockey team. The authors mention the injury paradox (lines 256-259). Unfortunately, they also emphasise personal protective equipment as solution. This suggestion is somewhat contrary to the general injury prevention literature, which tend to discourage too much onus on personal protective equipment at the expense of more effective strategies to reduce the risk of injury.
Minor issues:
The qualifier “high caliber” is rather uncommon in the sports literature. Consider using a more conventional qualifier to describe the level of sport. Furthermore, junior hockey is eligible for inclusion, but there is no indication of distinguishing between different levels of junior hockey. Were all studies on junior hockey included? Why make a distinction for adult hockey (i.e. international or professional only), but not junior hockey? The title, abstract, and introduction does not mention that this review is restricted men’s hockey. That fact is only mentioned in Table 1. This limitation of the review ought to be clearer at the outset. The flowchart in Figure 1 is not rendered properly (i.e. all the arrows are missing). Please use the PRISMA guideline flowchart. In Table 2, the column with the study title can be omitted. The section on injury definition (Section 2 in the Results) could be better structured. Present the three categories first. Then, make it clear that the examples of adding reportable injuries is only relevant in the context of Category 3 (time-loss) injuries. Similarly, the section on exposure (Section 3 in the Results) could be better structures. The denominators can be categorised in several ways: training versus competition, team versus individual (e.g. injuries per game versus injuries per athlete exposure), and exposure versus exposure-time (injuries per athlete-games versus injuries per athlete-game hours). Then, there is the issue about estimating the number of exposures or exposure-time versus measuring actual number of exposure or exposure-time. This distinction often pertains to whether exposure data is collected at the team level or individual level. Be consistent in using “number of” and “#” when discussion exposure definitions.
Author Response
Please see the attachment. Thank you very much for the thorough review.

Reviewer 2 Report
27) first sentence should be less emotive (reads too much like a magazine)
42) need to be more clear about region (e.g. country(ies)) and better define “high caliber” (e.g. junior level for many countries may not be considered high caliber)
83) what constitutes “professional” varies greatly between country and does professional simply mean athletes are “paid”? e.g. semi pro? For example, there is a big difference in money and ability between the NHL and U.S. minor league teams
- how many leagues are included in this? What is the “level of each of the se leagues? Need a lot more description How many athletes ?
84) collegiate systems are very different between Canada and the U.S. – for U.S. does this include NCAA, club, NAIA, junior college, etc.? need to be more detailed
85) need to define “junior level” very clearly (age, level, etc.)
Some of these questions are addressed in the very large table but not completely
I think a major problem with this study is that….
“Europe” is mostly represented by Sweden
Many of these studies only looked at specific injuries (e.g. ACL, hip) vs all injuries
“college” is too varied; D1, D3, college hockey?
Japan isn’t in Europe or the U.S. or Canada
What is the U.S. hockey league ?
Really no information about these athletes physical capabilities
I applaud the effort to better define injury and injury situation in hockey; however, I don’t believe there is enough data to conduct a review on this topic at this time
Author Response
Please see the attachment. Thank you very much for your review.

Reviewer 3 Report
Reviewing this article have been a pleasure. Apart from the interest of the topic, I consider it relevant in the current research connected to injuries in high performance Hockey. I would like to commend the authors on providing a well written and interesting research. The comments below are provided in order to enhance the current manuscript.
Abstract
As the main objective of this integrative literature review is to review and define “Low-Extremity” musculoskeletal injuries, I think that apart from as a key word, this term should be also during the abstract.
Introduction
There is a good analysis about the situation in the introduction, but the manuscript could be improved adding a more extended information about definitions and more updated injury models in other team sports, e.g. rugby (please justify with references).
Methods
Even though it is specified in the limitations of the study, I need to stress, that apart from PubMed, close databases like Medline, Embase or SportDiscus would provide wider view of the injury rate, definition and exposure in this sport.
Inclusion criteria “Published in English language”. Perhaps, the articles published in another language add some good information to this research. I think that the first criteria “Peer reviewed original articles” would assure the quality of the data and results obtained in different publications, despite the language.
Exclusion criteria in “Type of Injury” should exclude upper body musculoskeletal injuries in other to be more specific with the main goal of this research “Lower extremity musculoskeletal injury rates”. In the same criteria “anatomical location” appears in both, inclusion criteria and exclusion criteria.
Category 1 need further information.
“Exposure” as one of the goals of this review, could have been used within the search strategy as a key word.
Figure 1 Flow chart could show the order of the process
Discussion
There is a good discussion about the information gathered in the review.
Conclusions
Conclusions and future research direction are well explained.
Author Response
Please see attachment. Thank you very much for the thorough review.

Round 2
Reviewer 1 Report
Thank you for the opportunity to re-review the manuscript titled “What is Injury In Ice Hockey: An Integrative Literature Review on Injury Rates, Injury Definition, and Athlete Exposure in Men’s Elite Ice Hockey”. Although the authors have made several improvements to their manuscript, most of them are cosmetic changes addressing more trivial issues, while the more substantial weaknesses remain unaddressed. Please see specific comments below.
While I agree with the authors that lower limb MSK injuries in ice hockey deserves more attention than it has received to date, I think it is unwise to attempt to make a case for improved injury surveillance on that basis. Surveillance of lower limb MSK injuries should not occur in isolation from surveillance of other injuries in ice hockey. It would be counterproductive and a waste of resources to develop multiple injury surveillance methodologies targeting isolated injury problems in ice hockey. Why have develop one specifically for lower limb MSK? Would it not be more beneficial to have a consistent approach for lower and upper limb injuries? And for MSK and non-MSK injuries? A lack of consistency across injury types and body regions will only lead to reduced ability to compare the burden of injury across types and regions, thereby making priority-setting and resource allocation more difficult. The goal ought to be a more unified approach to facility intra- and cross-sport comparisons. Lower limb MSK injuries in ice hockey may be under-researched, but that is not a sufficient argument for developing specific surveillance methods. If the authors insist that specific surveillance methods are needed for lower limb MSK injuries in ice hockey, then it behoves them to explain why these injuries are so unique that they require a separate surveillance approach. Further to the point above, the authors response to Reviewer #1 (point 7) is a tacit admission a more general, comprehensive approach is needed when addressing methodological issues. That is, the authors state that their initial approach was to use global injuries when discussing injury and exposure definitions, while their discussion specific to lower limb MSK injuries only pertains to injury rates and distribution. The added ‘rationale’ is clearly retrofitted to rationalise the present version of the manuscript; it is not a strong rationale for why this review had to be this way. The authors response to Reviewer #1’s questions about why this review is restricted to men’s elite ice hockey is unsatisfactory. This criticism follows the same logic as point 1 above. If the goal is to improve surveillance methodology, then why restrict to men’s elite? What is so unique about lower limb MSK injuries in men’s elite ice hockey that it needs its own approach to surveillance? In their response to Reviewer #1’s comment about the search strategy, the authors are being either disingenuous or naïve. It is obvious that the search strategy is extremely restrictive. This is evidenced both by the fact that only 106 records were retrieved in PubMed, and by the fact two-thirds of included studies were identified through hand-searching. How many of the included studies identified through hand-searching were actually indexed in PubMed despite not being captured by the author’s original search? (I suspect most of them would be indexed in PubMed.) To suggest that this huge proportion of included studies identified through hand-searching is due to their choice of only using PubMed is untenable. This is evidenced by the fact that conducting a corresponding search in SPORTDiscus only identified one additional eligible study. (By the way, despite the authors’ assertion that the PubMed and SPORTDiscus search strings are equivalent, they are not. The PubMed search string includes the element ‘AND (definition OR rate OR rates)’, whereas the SPORTDiscus search string does not.) I strongly recommend revising the search string to decrease the reliance on hand-searching and potential for bias. The authors have attempted to describe why the Results and Discussion about injury rates pertain to MSK injuries of upper and lower limbs, while the Results and Discussion on specific anatomical areas pertain to MSK injuries of the lower limb only. I think this switch makes the manuscript confusing and unfocused, more consistency would be preferable. This manuscript is suffering from the authors trying to do too many disparate different things at the same once. The authors really ought to think about what their main objective is, and then align the manuscript accordingly. In their response to Reviewer #1’s comments, the authors attempt to defend their preference for using a time-loss definition. I agree with most of the points the authors mention (e.g. choice of injury definition should reflect the goals of the surveillance). What is lacking is a discussion of the limitations of choosing a time-loss definition. As well known, time-loss injuries represent the tip of the proverbial ice berg. It is also recognised that not all non-time-loss injuries are inconsequential, they can impair both individual and team performance. This is arguably of greater concern at the elite level where much more is often at stake. One of the common arguments for using a time-loss definition is that that it is easier to use, especially in resource-poor setting (e.g. community-level sport where there is limited or no access to team medical personnel). But this is generally not the case at the elite level, where resources for medical personnel and timely diagnoses of injuries is typically not an issue. Thus, I could argue that a broader injury definition is preferable at the elite level where injuries are of greater importance and the required resources are available. Time-loss injury surveillance is a god fall-back option, especially for resource-poor settings. This is also consistent with injury surveillance consensus statements in other sports such as rugby. Why would ice hockey be any different? The authors mention the injury paradox (lines 256-259). Unfortunately, they also emphasise personal protective equipment as solution. This suggestion is somewhat contrary to the general injury prevention literature, which tend to discourage too much onus on personal protective equipment at the expense of more effective strategies to reduce the risk of injury.
Reviewer 2 Report
I commend the authors on their responses and modifications.
Author Response
Thank you for your comments
Round 3
Reviewer 1 Report
Thank you for the opportunity to re-review the manuscript titled “What is Injury In Ice Hockey: An Integrative Literature Review on Injury Rates, Injury Definition, and Athlete Exposure in Men’s Elite Ice Hockey”. Although the authors have satisfactorily addressed all my previous comments, I would like to seek clarification on one point arising from the authors’ responses. The authors state that their new search “resulted in an additional 11 additional papers for full review”. It is unclear how that can be given that the number of full text articles assessed for eligibility in the new version is 24, compared to 27 in the previous version? More importantly, the authors state that they have not yet been able to obtain full text versions of 4 articles. The PRISMA flow chart does not clearly account for these 4 articles. Have they been excluded prior to full text assessment? That is, are they among the n=2188 articles discarded during the screening phase? If so, I think at the very least they should be accounted for among the n=24 full text articles assessed for eligibility. Secondly, the authors ought to wait for the interlibrary loan requests to be returned before finalising their manuscript, and then account for them in among either n=0 excluded articles or n=24 included articles, as appropriate. If the articles cannot be retrieved, then they should be accounted for among the n=0 excluded articles and clearly labelled as a failure to retrieve full text. I also recommend attempting to retrieve the missing articles directly from the corresponding authors via email.
Author Response
Please see attached. Thank you for your time and expertise
